# Cloning, Characterization, and RNA Interference Effect of the *UDP-N-Acetylglucosamine Pyrophosphorylase* Gene in *Cnaphalocrocis medinalis*

**DOI:** 10.3390/genes12040464

**Published:** 2021-03-24

**Authors:** Yuan-Jin Zhou, Juan Du, Shang-Wei Li, Muhammad Shakeel, Jia-Jing Li, Xiao-Gui Meng

**Affiliations:** Guizhou Provincial Key Laboratory for Agricultural Pest Management of Mountainous Regions, Institute of Entomology, Guizhou University, Guiyang 550025, China; zyj18786607036@163.com (Y.-J.Z.); juandudj@163.com (J.D.); Shakeelagri1947@gmail.com (M.S.); jjli0315@163.com (J.-J.L.); mengxiaogui1010@163.com (X.-G.M.)

**Keywords:** *Cnaphalocrocis medinalis*, UDP-N-acetylglucosamine pyrophosphorylase, gene cloning, expression pattern, RNA interference

## Abstract

The rice leaf folder, *Cnaphalocrocis medinalis* is a major pest of rice and is difficult to control. UDP-N-acetylglucosamine pyrophosphorylase (UAP) is a key enzyme in the chitin synthesis pathway in insects. In this study, the *UAP* gene from *C. medinalis* (*CmUAP*) was cloned and characterized. The cDNA of *CmUAP* is 1788 bp in length, containing an open reading frame of 1464 nucleotides that encodes 487 amino acids. Homology and phylogenetic analyses of the predicted protein indicated that CmUAP shared 91.79%, 87.89%, and 82.75% identities with UAPs of *Glyphodes pyloalis*, *Ostrinia furnacalis*, and *Heortia vitessoides*, respectively. Expression pattern analyses by droplet digital PCR demonstrated that *CmUAP* was expressed at all developmental stages and in 12 tissues of *C. medinalis* adults. Silencing of *CmUAP* by injection of double-stranded RNA specific to *CmUAP* caused death, slow growth, reduced feeding and excretion, and weight loss in *C. medinalis* larvae; meanwhile, severe developmental disorders were observed. The findings suggest that *CmUAP* is essential for the growth and development of *C. medinalis*, and that targeting the *CmUAP* gene through RNAi technology can be used for biological control of this insect.

## 1. Introduction

The rice leaf folder, *Cnaphalocrocis medinalis* (Lepidoptera: Pyralidae) is a polyphagous insect pest that attacks several species of plants, especially grasses (deccan grass, johnson grass, cockspur grass, couch grass, and rice) [1]. *C. medinalis* has been reported as one of the most severe pests of rice crops worldwide [2], and serious infestations have been documented in many countries in Asia, including India, Korea, Japan, China, Malaysia, Sri Lanka, and Vietnam [3]. *C. medinalis* undergoes complete metamorphosis, which consists of four developmental stages, i.e., egg, larva, pupa, and adult [4]. Rice crop damage by *C. medinalis* is caused by rolling the leaves longitudinally and scraping the mesophyll contents of fresh leaves [5,6]. During a severe infestation, plant growth is stunted due to a lack of photosynthesis, resulting in heavy yield losses [6]. In some instances, approximately 63% to 80% grain losses have been recorded during severe attack [7]. Currently, the control of *C. medinalis* relies mainly on chemical synthetic insecticides [8]. However, frequent application of synthetic insecticides leads to resistant populations of *C medinalis*, pollutes the environment, increases production cost, and brings about a threat to human health [9,10]. Resistance against chemical insecticides has been reported in *C. medinalis* throughout the world [4]. Therefore, there is an urgent need to find more efficient and safe methods to control this insect pest.

RNA interference (RNAi) is a post-transcriptional mechanism of gene knockdown that targets the mRNA of a desired gene [11,12], and was initially discovered in *Caenorhabtidis elegans* [13]. Owing to its specificity, RNAi has been successful as a gene-silencing tool in several insect orders such as Hymenoptera, Diptera, Coleoptera, Hemiptera, and Lepidoptera [14,15,16,17,18]. The delivery methods and selection of target genes are essential for effective RNAi-based insect-pest management [19]. The injection of double-stranded RNA (dsRNA) in the body of insect and orally feeding dsRNA have been successful in many species of insects [17,20,21]. Microinjections have been successfully used in several insect species, such as *Nilaparvata lugens* and *Tribolium castaneum* [22,23,24,25], and RNAi by injection has become an effective method for pest control.

Chitin biosynthesis begins with trehalose and involves at least eight critical enzymes in insects. UDP-N-acetylglucosamine pyrophosphorylase (UAP) is the seventh enzyme in the cascade [26,27], and catalyzes the transfer of a UMP moiety from UTP to N-acetylglucosamine to form UDP-N-acetylglucosamine, a substrate for chitin synthase (CHS). The biosynthetic pathway of chitin mainly consists of three subreactions. The first set leads to the formation of the amino sugar, GlcNAc, the second to its activated form UDP-GlcNAc, and the last yields the polymeric chitin from the amino sugar. The rate-limiting enzyme in the first subreaction appears to be glutamine-fructose-6-phosphate aminotransferase (GFAT). The critical enzyme in the second subreaction is UDP-N-acetylglucosamine pyrophosphorylase, and the key enzyme in the last subreaction is CHS. These three enzymes are major regulatory enzymes in the insect chitin synthesis pathway [28]. In general, most insects have only one *UAP* gene, except for *T. castaneum*, *Locusta migratoria*, and *L. decemlineata*, which have two *UAP* genes [29,30,31]. RNAi experiments demonstrated that knockdown of *TcUAP1* in *T. castaneum* or *LdUAP2* in *L. decemlineata* resulted in a reduction of chitin in the peritrophic matrix (PM), and thus affected the PM integrity and blocked insect development. RNAi targeted *LmUAP1* was also associated with higher mortality in *L. migratoria* [29]. Although there are some studies on the chitin synthesis pathway in insects, most studies focus on trehalose synthase and chitin synthase. Up to now, *UAP* in *C. medinalis* (*CmUAP*) has not been characterized.

In this present study, the *CmUAP* gene from *C. medinalis* was cloned and the expression pattern was determined using droplet digital PCR (ddPCR). Furthermore, the effect of *CmUAP* on the growth and development of *C. medinalis* was detected by using RNAi to reduce the gene expression. This study will assist in establishing a new target site for pest control and provide a theoretical basis for biological control by RNAi.

## 2. Materials and Methods

### 2.1. Insect Rearing and Sample Preparation

The larvae of *C. medinalis* were initially collected from rice fields in Guiyang, Guizhou, China (located at 106.735° E, 26.412° N). *C. medinalis* were raised on rice seedlings in the insectary of the Institute of Entomology at 26 ± 1 °C and 75 ± 5% relative humidity under a 14 h: 10 h light: dark photoperiod. After three consecutive generations, insects in the same developmental stage were used as the test samples for this experiment. Research samples included four life stages, i.e., the egg, first-fifth-instar larvae, pupa, and adult. Various tissues were dissected from adults, including the head, cuticle, epidermis, muscle, fat body, midgut, malpighian tubule, testis, ovary, hemolymph, wing, and leg. These samples were stored in RNAlater (Qiagen, Duesseldorf, Germany) at −80 °C until use.

### 2.2. RNA Extraction and cDNA Synthesis

Total RNA was extracted from different developmental stages and various tissues of *C. medinalis* adults rrusing the HP Total RNA kit (Omega Bio-Tek, Norcross, GA, USA) according to the instructions of the manufacturer. The quality and purity of the RNA were determined by agarose gel electrophoresis and a NanoDrop 2000 spectrophotometer (Thermo Fisher, Waltham, MA, USA), respectively. All samples were diluted with DEPC-treated water and stored at −80 °C until required. First-strand cDNA was synthesized using a RevertAid First Strand cDNA Synthesis Kit (Thermo Fisher, Waltham, MA, USA) according to the manufacturer’s instructions and then stored at −20 °C until use.

### 2.3. Cloning CmUAP by Both RT-PCR and RACE

For the reverse transcription-polymerase chain reaction (RT-PCR), the specific primers were designed using Primer Premier 6.0 (PREMIER Biosoft, San Francisco, CA, USA) based on the fragments of the *C. medinalis* transcriptome [32]. RT-PCR was performed in a 40-μL reaction system consisting of 3 μL of cDNA, 1 μL each of forward and reversed primers (20 μM each), 8 μL of dNTP mixture (2.5 mM each), 4 μL of 10× LA PCR buffer II, 0.5 μL of LA Taq DNA polymerase (5 U/μL), and 22.5 μL of sterile water, in a C1000 Thermal Cycler (Bio-Rad, Hercules, CA, USA). The reaction parameters were as follows: pre-denaturation at 94 °C for 1 min; 30 cycles of denaturing at 94 °C for 30 s, annealing at 53 °C for 30 s, and extension at 72 °C for 2 min; and a final extension at 72 °C for 5 min. PCR products were examined by electrophoresis and then purified using an Agarose Gel DNA Extraction Kit (Sangon Biotech, Shanghai, China). The purified product was submitted to Sangon Biotech Company (Shanghai, China) for sequencing.

The full-length cDNA of *CmUAP* was obtained by using the rapid amplification of cDNA ends-PCR (RACE-PCR). The 3′ end of this cDNA was amplified using the SMARTer RACE 3′ Kit (Clontech Laboratories, Mountain View, CA, USA). The 3′-RACR was used to amplify the 3′ end with two sets of specific nested primers, namely, CmUAP-3F1 and CmUAP-3R1 as well as CmUAP-3F2 and CmUAP-3R2. The conditions for the first-round PCR were: pre-denaturation at 94 °C for 1 min; 30 cycles of denaturing at 94 °C for 30 s, annealing at 53 °C for 30 s, and extension at 72 °C for 50 s; and a final extension at 72 °C for 5 min. The parameters of the second-round reaction were the same as those of the first round, except that an annealing temperature of 56 °C was used. The 3′-RACE products were purified and sequenced, and then these sequences from both RT-PCR and RACE were compared with the NCBI Nt database using BLAST to determine whether they were fragments of an *UAP* gene. The full-length cDNA of *CmUAP* was obtained by overlapping the two correct fragments. The list of primers used in this study is shown in Table 1.

### 2.4. Bioinformatic Analyses of CmUAP

We used ORFfinder to search the open reading frame (ORF) of *CmUAP* (https://www.ncbi.nlm.nih.gov/orffinder, accessed on 5 May 2020). ProtParam (https://web.expasy.org/protparam, accessed on 5 May 2020) was used to analyze the molecular weight and isoelectric point (pI) of the zymoprotein. The signal peptide was predicted with SignalP 5.0 (https://services.healthtech.dtu.dk/service.php?SignalP-5.0, accessed on 6 May 2020), and the transmembrane helices were predicted using TMHMM (https://services.healthtech.dtu.dk/service.php?TMHMM-2.0, accessed on 6 May 2020). Motifs and domains were analyzed using ScanProsite (https://prosite.expasy.org/scanprosite, accessed on 7 May 2020) and InterPro (http://www.ebi.ac.uk/interpro, accessed on 8 May 2020), respectively. Phosphorylation sites were expected with NetPhos (https://services.healthtech.dtu.dk/service.php?NetPhos-3.1, accessed on 11 January 2021). A phylogenetic tree was constructed by MEGA X based on the UAP amino acids of 26 insect species using the neighbor-joining (NJ) method. The three-dimensional (3D) structure of CmUAP was predicted by homology modeling in SWISS-MODEL (https://swissmodel.expasy.org, accessed on 22 February 2021), and then visualized using PyMOL 2.4 (Schrodinger, New York, NY, USA).

### 2.5. Gene Expression Analyses Using ddPCR

The mRNA expression levels of *CmUAP* were detected by droplet digital PCR (ddPCR) at different developmental stages and in various tissues of adults. The disposable eight-channel DG8 cartridge was placed in the cartridge holder and 20 μL of PCR mixture was transferred to the middle well of the cartridge. The bottom well was filled with 70 μL of droplet generation oil. The cartridge containing the PCR mixture and the oil was placed into the QX200 Droplet Generator (Bio-Rad, Hercules, CA, USA). Then, 40 μL of droplets from the cartridge were transferred into a 96-well PCR plate. The plate was heat sealed at 180 °C for 5 s with a permeable foil using a PX1 PCR plate sealer (Bio-Rad, Hercules, CA, USA). Subsequently, PCR amplification was performed in a C1000 Touch Thermal Cycler with 96-deep well reaction module, and the reaction system and conditions are listed in Table 2. The 96-well PCR plate was cooled at room temperature at the end of the reaction. The PCR plate containing the amplified droplets was placed in a QX200 Droplet Reader (Bio-Rad, Hercules, CA, USA) to count positive and negative droplets. Three replicates were performed for each sample. Data were analyzed using QuantaSoft software (Bio-Rad, Hercules, CA, USA) and SPSS 22.0 (SPSS Inc., Chicago, IL, USA).

### 2.6. RNA Interference

Two online RNAi design tools, i.e., siRNA at Whitehead (http://sirna.wi.mit.edu, accessed on 5 May 2020) and siDirect (http://sidirect2.rnai.jp, accessed on 5 May 2020) were used to search for fragments targeted *CmUAP* mRNA. The *CmUAP* cDNA was used as the template and amplified using CmUAP-iF and CmUAP-iR primers. After purification, the PCR products were inserted into pMD 18-T vector (Takara Bio, Dalian, China) for sequencing. Clones containing the correct sequences were cultured. Plasmids were extracted and used as templates to amplify target fragments with CmUAP-dsF and CmUAP-dsR primers. After purification of PCR products, high concentrations (not less than 300 ng/μL) of DNA were obtained. Ds*CmUAP* was synthesized using a TranscriptAid T7 High Yield Transcription Kit (Thermo Fisher, Waltham, MA, USA). Briefly, the in vitro transcription system (20 μL) contained 2 μL of nuclease-free water, 4 μL of 5× reaction buffer, 8 μL of ATP/CTP/GTP/UTP mix, 4 μL of DNA template (with T7 at both ends), and 2 μL of enzyme mix. After vortexing and brief spinning, the mixture was incubated at 37 °C for 8 h. Ds*CmUAP* was purified using a GeneJET RNA Purification Kit (Thermo Fisher, Waltham, MA, USA). The integrity and quality were evaluated with agarose gel electrophoresis and a NanoDrop 2000 spectrophotometer (Thermo Fisher, Waltham, MA, USA). The same method was used to prepare ds*GFP* for the control group.

For the injection, 20 third-instar larvae with consistent growth and development were selected as test insects. The eighth abdominal segment of each larva was selected as the injection site. One microliter of ds*CmUAP* or ds*GFP* (2 μg/μL) was injected into each larva in the direction of blood flow using a Microinjection Syringe Pump (World Precision Instruments, Sarasota, FL, USA). These injected larvae were placed onto fresh rice leaves in glass tubes and reared in an artificial climate chamber. The rice leaves were replaced with fresh ones every two days. The larvae were observed for feeding, phenotype, and survival; the larvae and pupae were weighed at a 24-h interval. Excreta were collected, dried, and weighed. At pupal and adult stages, expression level of the *CmUAP* gene was detected every 24 h using ddPCR. RNAi efficiency = [(*CmUAP* expression levels in the control group − *CmUAP* expression levels in the experimental group)/*CmUAP* expression levels in the control group] × 100%. Four replicates were performed for each of experimental and control groups.

### 2.7. Statistical Analysis

Data were expressed as the mean ± SD from at least three independent experiments. Statistical analysis was carried out using one-way analysis of variance (ANOVA) followed by Duncan’s multiple range test with SPSS 22.0 (SPSS Inc., Chicago, IL, USA). A *P*-value less than 0.05 was considered statistically significant.

## 3. Results

### 3.1. Characteristic Analyses of CmUAP and the Zymoprotein

The full-length cDNA of *CmUAP* was 1788 bp in length, containing an ORF of 1464 nucleotides that encoded 487 amino acids (Figure 1). The molecular formula of CmUAP protein was C_7892_H_12312_N_2066_O_2244_S_95_, with a molecular weight of 54.5 kDa and a theoretical isoelectric point of 5.76. The signal peptide and transmembrane structure were not found in this zymoprotein. The protein was predicted to possess a total of 38 potential phosphorylation sites, including 24 serine phosphorylation sites, nine threonine phosphorylation sites, and five tyrosine phosphorylation sites. Homology modeling revealed that the three-dimensional molecular structure of CmUAP formed a homodimer containing 37 α-helices, 37 β-pleated sheets, and 71 random coils (Figure 2).

### 3.2. Homology Comparison and Cluster Dendrogram

Based on a BLAST search against the NCBI Nr database, CmUAP showed 91.79%, 87.89%, and 82.75% similarity with UAPs of *Glyphodes pyloalis*, *O. furnacalis*, and *Heortia vitessoides*, respectively. Additionally, the similarity of CmUAP with UAPs of more than 30 species of lepidopterans exceeded 70%, suggesting the relatively conserved evolutionary characteristic of this protein in lepidoptera. The phylogenetic tree of UAPs from 26 lepidopteran species showed that CmUAP clustered into a clade with UAPs from *G. pyloalis* and *O. furnacalis* (Figure 3). These results indicated that RLF was the closest relative of these two insects.

### 3.3. Gene Expression Profiles

The ddPCR results showed that *CmUAP* was expressed throughout the developmental stages of *C. medinalis*, with the highest expression level in adults and the lowest level in eggs (Figure 4). The expression level of *CmUAP* in adults was five times higher than that in eggs. *CmUAP* was expressed at different levels in the tested 12 adult tissues (Figure 5). This gene was expressed at the highest level in the midgut, followed by the epidermis and cuticle, and at the lowest level in the wings. The *CmUAP* expression level in the midgut was 21 times higher than that in the wings.

### 3.4. RNAi Effects of CmUAP

Ds*CmUAP* caused significant reductions in *CmUAP* expression levels at the larval, pupal, and adult stages (Figure 6). RNAi efficiency reached a maximum of 68.6% at the third day and then gradually decreased over time. RNAi efficiency in the larva was much higher than that in the adult, and the efficiency rose in the pupa compared with that in the fifth-instar larva and adult (Figure 7). The corrected mortality rate of ds*CmUAP*-injected larvae (43%) was about three times that of dsGFP-injected larvae (15%) at the seventh day post injection (Figure 8). At the same time, RNAi gave rise to significant phenotypic changes of *C. medinalis*, including reduced excretion, smaller body size and weight loss (Table 3), and deformity. Some experimental individuals injected with ds*CmUAP* showed disrupted molting and abnormal development, which was manifested by the enlarged head (Figure 9a,b) and smaller abdomen (Figure 9c) with ulceration on the body surface (Figure 9d,e). In addition, disrupted molting and abnormal body color at the larval stage led to pupation failure (Figure 9f,g). In contrast, no significant phenotypic changes were observed in the control. At the pupal stage, some ds*CmUAP*-injected individuals failed to pupate, but showed abnormal phenotypes, such as incomplete pupa, darkening pupa, and pupa without shell (Figure 10). Furthermore, ds*CmUAP* brought about a prolonged pupal stage and pupal death within 3–4 days post injection. Moreover, RNAi targeted ds*CmUAP* inhibited adult emergence, shortened adult life span, reduced egg production, and caused fewer egg hatche (Table 3). RNAi also caused deformities of adult wings, such as abnormal wing folding and unfolding, wing curling, and wings failing to cover the abdomen (Figure 11). Deformed individuals lost the ability to fly but survival, mating, and reproduction were not affected. These results indicated that silencing of *CmUAP* caused severe developmental disorder and death of *C. medinalis*.

## 4. Discussion

Chitin is the second most crucial biological polysaccharide in nature after cellulose. It is widely distributed, not only in arachnids, crustaceans, and insects but also in lower invertebrates such as sponges, coelenterates, nematodes, and mollusks [33]. In insects, chitin is an essential part of the body wall or cuticle, peritrophic matrix, salivary glands, eggshells, trachea, and the sites of muscle attachment [28]. Chitin also plays a key role in growth and development and in the molting process of insects [27,34]. Therefore, the balance between degradation of old cuticle and formation of a new one, which is stimulated by various enzymes, is crucial for growth and molting in insects [35]. UAP is a critical regulatory enzyme in the insect chitin synthesis pathway. UDP-G1cNAc is produced by UAP for the biosynthesis of O- or N-linked oligosaccharides and the formation of GPI (glucose-6-phosphate isomerase) anchors, which is involved in the formation of cell wall peptidoglycan in bacteria. UAP also acts a precursor substance of chitin synthase in eukaryotic species and in the composition of cell wall in fungi. Meanwhile, UAP plays an important role in the physiological metabolism of insect cells [36,37]. Most studies on key enzymes in the chitin synthesis pathway of insects have focused on chitin synthase and trehalase.

By comparing with other species, we found that CmUAP is highly related to UAPs of other lepidopterans and is highly conserved in this order. Therefore, the *UAP* gene can be used as a target to develop broad-spectrum insecticides against lepidopterous pests. The expression pattern and characteristics of *CmUAP* were consistent with those of other genes in the chitin synthesis pathway of *C. medinalis*, such as *CmCHSB* (*chitin synthase B*) and *CmHK* (*hexokinase*) genes. Higher expression levels were observed in adults, followed by second-fourth-instar larvae as compared to other larval stages and eggs, which was similar to *CmCHSB* [38]. Moreover, the high expression of *CmUAP* was observed in the midgut and epidermis of *C. medinalis*; this may be related to functions of nutrient absorption as well as growth and development.

RNAi targeted *CmUAP* resulted in low expression of this gene in *C. medinali*, and pupae and adults exhibited abnormal development; this phenomenon was consistent with that after *CmHK* interference [39]. The same phenomenon was also confirmed in *S. exigua* [40]. In addition, silencing of *TcUAP1* in *T. castaneum* caused developmental retardation and death of larvae [31], which is consistent with the phenotype that we observed at the larval stage in *C. medinali* post ds*CmUAP* injection. After silencing of *CmUAP*, we found a significant reduction in body weight of larvae and pupae, likely due to a decrease in UAP regulation that resulted in insufficient chitin synthesis, affecting nutrient uptake by insects and the formation of the integument. 

During pupal formation in *C. medinalis*, silencing of *CmUAP* caused malformation and pupation failure as well as prolonged pupal stage and reduced pupal weight. These findings were similar to observations in *S. exigua*, in which silencing of *UAP* caused decreased gene expression, emergence of malformed pupae, and a long duration of pupation [41]. These data suggest that *UAP* is essential for the pupal formation in lepidopteran.

At the adult stage, we observed that ds*CmUAP* resulted in deformities, shortened lifespan, and a significant reduction of egg laying and hatching rate. The adult deformities were mainly in the form of wing deformation, which has not been reported before. The presumed reduction in the production of chitin likely contributed to impaired wing development, reduced lifespan, and egg production. RNAi efficiency during the pupal stage was as high as 53.4%, suggesting that *CmUAP* played a crucial role during the transition from the pupa to the adult. We speculated that due to the low expression of *CmUAP* and reduced synthesis of chitin in the larva, this insect did not have sufficient chitin at the pupal and adult stages, resulting in incomplete and malformed wing development of *C. medinalis* adult.

In the present study, we observed that RNAi targeted *CmUAP* resulted in aberrations and death at various stages of *C. medinalis*. More research is needed to determine if the reduction in *UAP* transcripts will result in reduced catalysis of GlcNAc-1-P + UTP into UDP-GlcNAc + PPi and thus decrease synthesis of UDP-GlcNAc. In our future work, we will evaluate RNAi effect of ds*CmUAP* by oral administration on *C. medinalis* and consider transferring the dsRNA into rice using transgenic technology for more effective control of *C. medinalis*.

## 5. Conclusions

In this study, the *Cm*UAP gene was cloned and characterized from *C. medinalis*. Knockdown of *CmUAP* in *C. medinalis* using RNAi caused severe developmental disorders, malformation, and death at various developmental stages, which demonstrated the important role of *CmUAP* in the development of this pest. These results provide the basis for further studies on the properties and functions of *CmUAP*, as well as the development of new control strategies using RNAi for this damaging pest.

## Figures and Tables

**Figure 1 genes-12-00464-f001:**
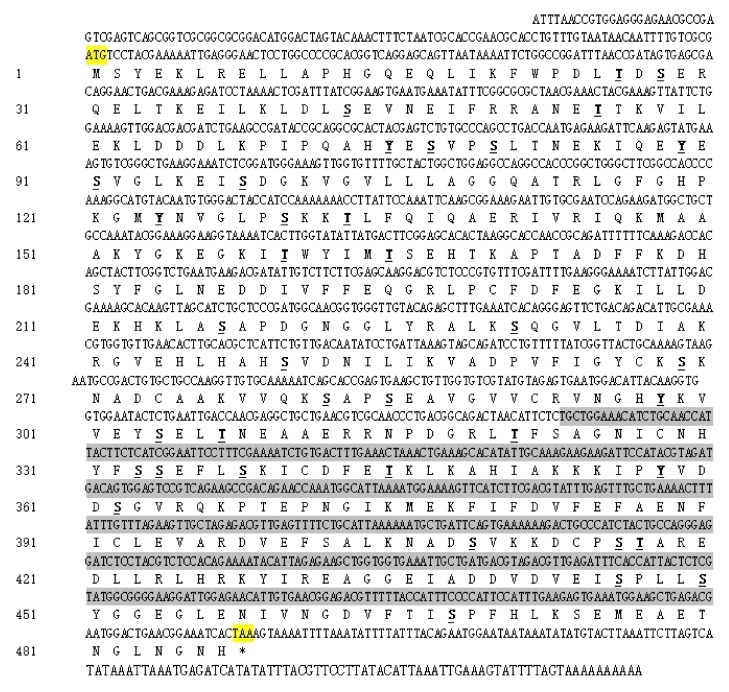
Nucleotide and deduced amino acid sequence of *CmUAP* cDNA. The start codon and stop codon are marked in yellow, the phosphorylation sites are marked in underlined bold font, and target sequence for RNAi is shaded.

**Figure 2 genes-12-00464-f002:**
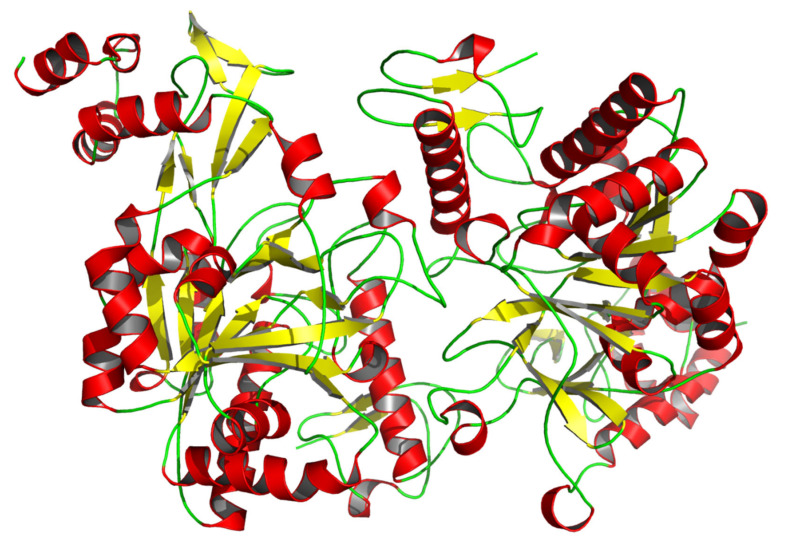
Three-dimensional molecular structure of CmUAP. This graphic was drawn using PyMOL 2.4 based on the CmUAP.pdb data. Red represents α-helices, yellow represents β-pleated sheets, and green represents random coils.

**Figure 3 genes-12-00464-f003:**
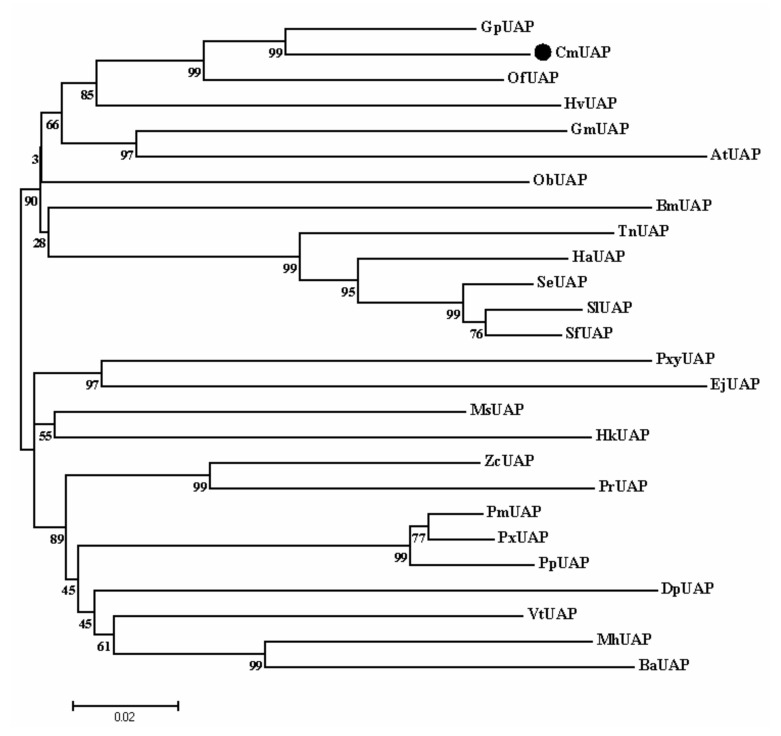
A cluster dendrogram of UAPs from 26 lepidopterans. The diagram was constructed using the neighbor-joining method (NJ) in MEGA X with 1000 replications. The bootstrap confidence values are indicated at the nodes. CmUAP is labeled with a black circle. Different lepidopteran species and corresponding GenBank accession numbers are indicated below. PmUAP from *Papilio Machaon* (XP_014356302.1), PxUAP from *Papilio xuthus* (XP_013163486.1), HaUAP from *Helicoverpa armigera* (XP_021195694.1), ZcUAP from *Zerene cesonia* (XP_038220429.1), PpUAP from *Papilio polytes* (XP_013139346.1), SeUAP from *Spodoptera exigua* (ACN29686.1), SlUAP from *Spodoptera litura* (XP_022831608.1), BmUAP from *Bombyx mandarina* (XP_028040171.1), VtUAP from *Vanessa tameamea* (XP_026500069.1), GmUAP from *Galleria mellonella* (XP_026759480.1), DpUAP from *Danaus plexippus* (XP_032528285.1), MhUAP from *Maniola hyperantus* (XP_034838984.1), TnUAP from *Trichoplusia ni* (XP_026725491.1), SfUAP from *Spodoptera frugiperda* (XP_035457283.1), PxyUAP from *Plutella xylostella* (XP_037972326.1), BaUAP from *Bicyclus anynana* (XP_023949028.1), PrUAP from *Pieris rapae* (XP_022127843.1), AtUAP from *Amyelois transitella* (XP_013189001.1), HkUAP from *Hyposmocoma kahamanoa* (XP_026317391.1), EjUAP from *Eumeta japonica* (GBP41850.1).

**Figure 4 genes-12-00464-f004:**
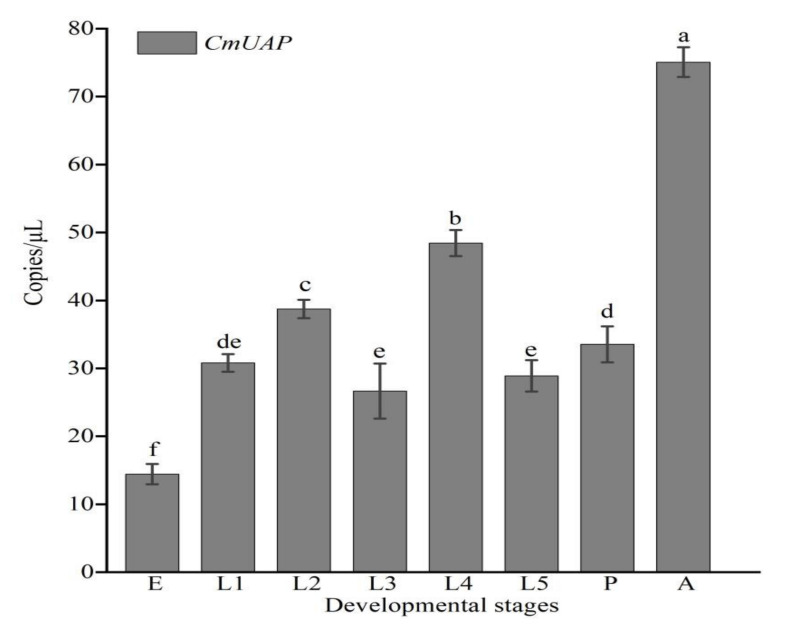
Expression levels of *CmUAP* at different developmental stages. E, Egg; L1–L5, first-fifth-instar larvae; P, Pupa; A, Adult. Each bar represents the mean ± SD. Different letters above the bars indicate significant differences at *p* < 0.05 based on Duncan’s test.

**Figure 5 genes-12-00464-f005:**
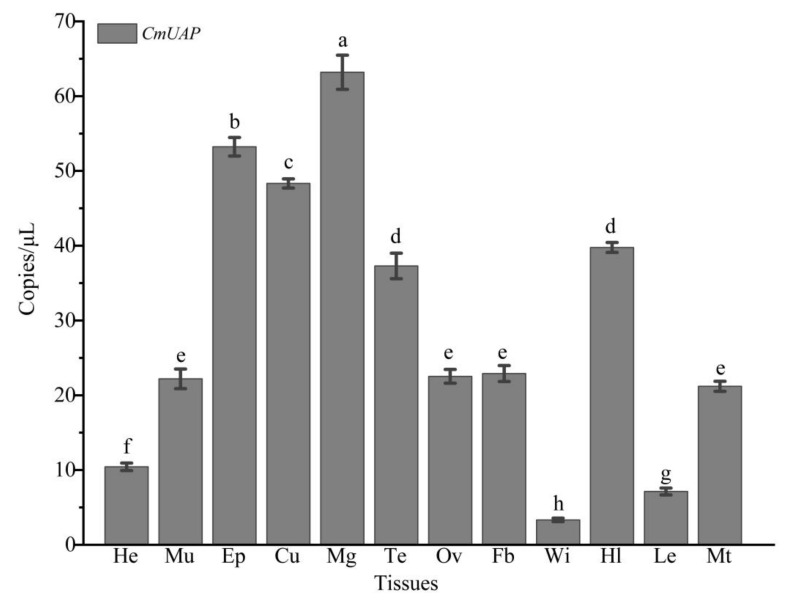
Expression levels of *CmUAP* in various tissues of *C. medinalis* adults. He, Head; Mu, Muscle; Ep, Epidermis; Cu, Cuticle; Mg, Midgut; Te, Testis; Ov, Ovary; Fb, Fat body; Wi, Wing; Hl, Hemolymph; Le, leg; Mt, Malpighian tubule. Each bar represents the mean ± SD. Different letters above bars indicate significant differences at *p* < 0.05 based on Duncan’s test.

**Figure 6 genes-12-00464-f006:**
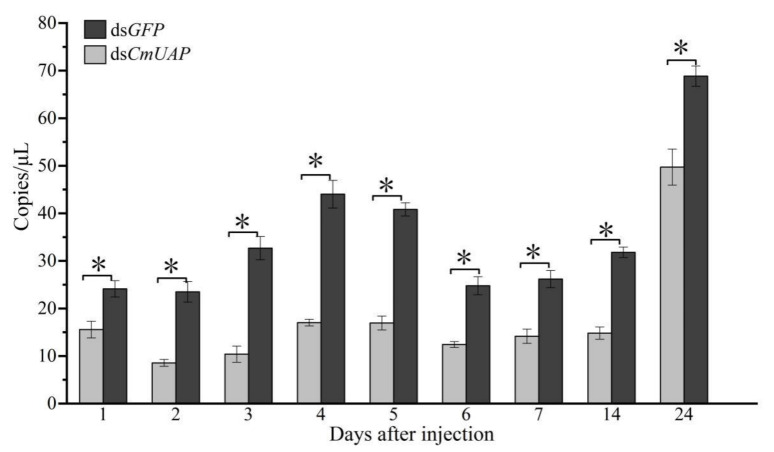
Expression levels of *CmUAP* at different times after dsRNA injection. 1–7, *CmUAP* expression in larvae 1–7 days post injection; 14, *CmUAP* expression in the pupa 14 days post injection; 24, *CmUAP* expression in the adult 24 days post injection. Each bar represents the mean ± SD. Asterisks above bars indicate significant differences at *p* < 0.05 according to Duncan’s test.

**Figure 7 genes-12-00464-f007:**
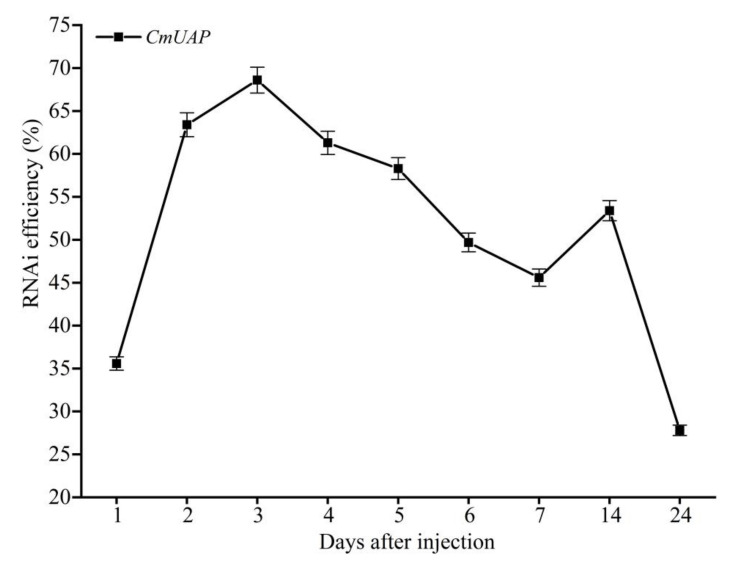
RNAi efficiency at different times. 1–7, RNAi effect on larvae 1–7 days post injection; 14, RNAi effect on the pupa 14 days post injection; 24, RNAi effect on the adult 24 days post injection.

**Figure 8 genes-12-00464-f008:**
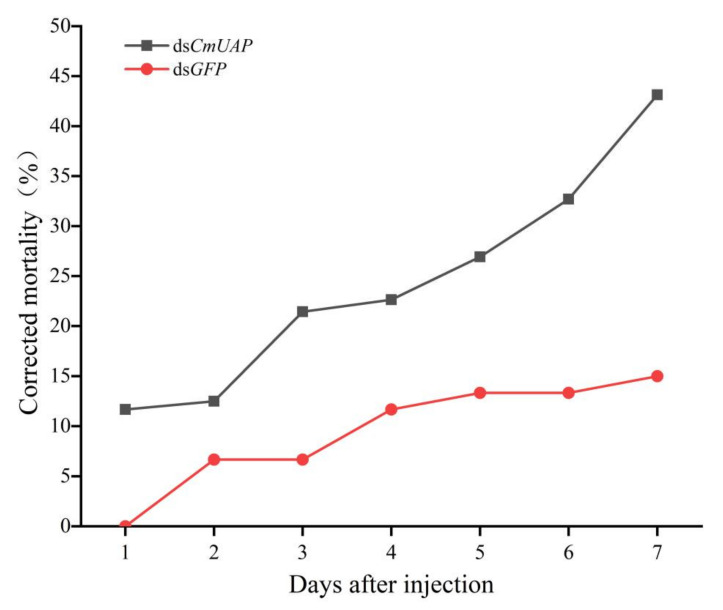
Corrected mortality of larvae within 7 d after dsRNA injection. Different letters above the broken lines indicate significant differences between treatments on the same day (*p* < 0.05, Duncan’s test).

**Figure 9 genes-12-00464-f009:**
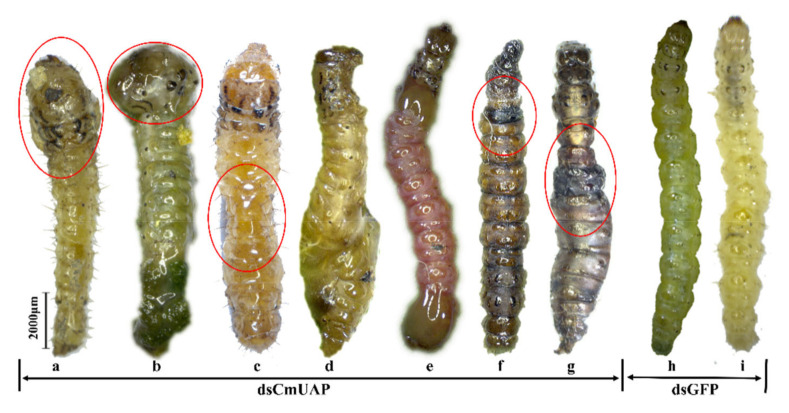
Abnormal morphology of *C. medinalis* larvae after dsRNA injection. (**a**–**g**), phenotypes of larvae after ds*CmUAP* injection; (**h**) and (**i**), ds*GFP*-injected larvae. Red circles indicate the phenotypic changes of the larvae. Scale bar represents 2000 μm.

**Figure 10 genes-12-00464-f010:**
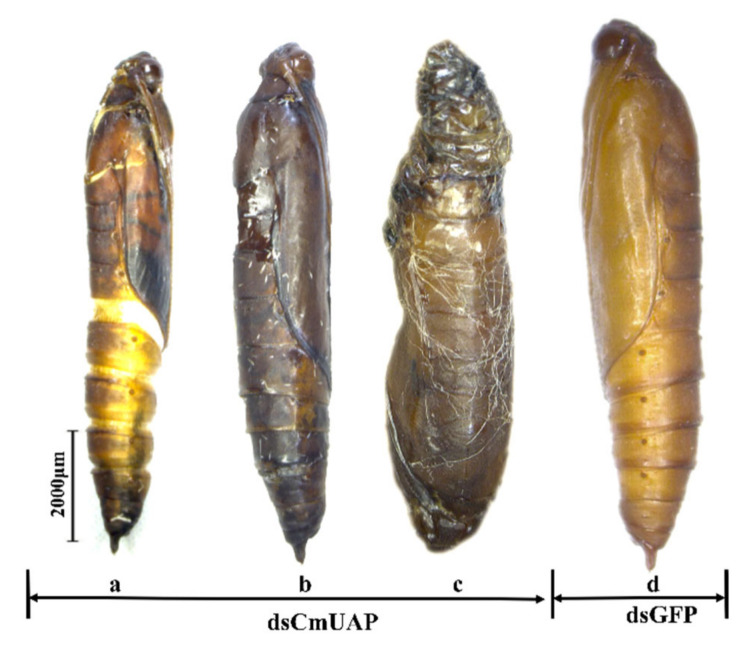
Abnormal morphology of *C. medinalis* pupae after dsRNA injection. (**a**), incomplete pupation; (**b**), darkening pupa; (**c**), no pupal shell formation; (**d**), ds*GFP*-injected pupa. Scale bar represents 2000 μm.

**Figure 11 genes-12-00464-f011:**
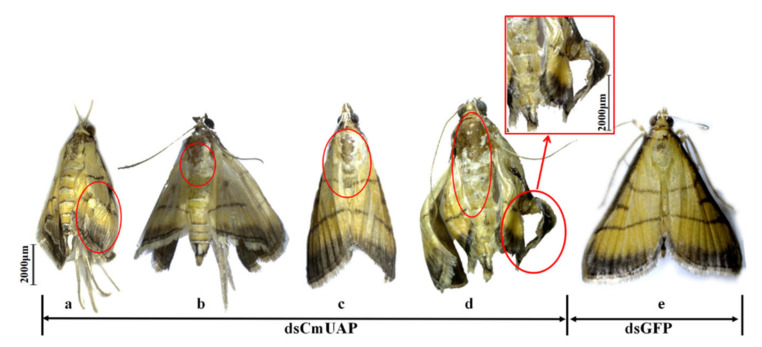
Abnormal morphology of *C. medinalis* adults after dsRNA injection. (**a**), abnormal wing folding; (**b**), abnormal wing unfolding; (**c**), wings failing to cover the abdomen; (**d**), wing curling; (**e**), ds*GFP*-injected adult. Malformed phenotypes are marked with red circles. Scale bar represents 2000 μm.

**Table 1 genes-12-00464-t001:** Primers used for cloning and expression analysis of *CmUAP* from *C. medinalis*

Primer Name	Primer Sequence (5′ → 3′)	Primer Usage
CmUAP-F	ATTTAACCGTGGAGGGAGAAC	RT-PCR
CmUAP-R	GTGATTTCCGTTCAGTCCATTC	
CmUAP-3F1	ATACGTAGATGACAGTGGAGTC	RACE-PCR
CmUAP-3R1	GCTGTCAACGATACGCTACGTAAC	
CmUAP-3F2	TCAGAAGCCGACAGAACCAAAT	
CmUAP-3R2	GCTACGTAACGGCATGACAGTG	
CmUAP-dF	CGGATGGGAAAGTTGGTGTTT	ddPCR
CmUAP-dR	TCTTCTGGATTCGCACAATTCT	
CmUAP-iF	TGGAAACATCTGCAACCATTACTTC	dsRNA synthesis
CmUAP-iR	CGAGAGTAATGGTGAAATCTCAACG	
CmUAP-dsF	taatacgactcactatagggTGGAAACATCTGCAACCATTACTTC	
CmUAP-dsR	taatacgactcactatagggCGAGAGTAATGGTGAAATCTCAACG	
GFP-iF	GCCAACACTTGTCACTACTT	
GFP-iR	GGAGTATTTTGTTGATAATGGTCTG	
GFP-dsF	taatacgactcactatagggGCCAACACTTGTCACTACTT	
GFP-dsR	taatacgactcactatagggGGAGTATTTTGTTGATAATGGTCTG	

Note: the lowercase letters in the primers represent the sequence of the T7 promoter.

**Table 2 genes-12-00464-t002:** The reaction system and procedure of droplet digital PCR

Component	Volume Per Reaction, μL	Final Concentration	Cycling Step	Temperature, °C	Time	Number of Cycles	Other Reaction Conditions
2 × QX200 ddPCR EvaGreen Supermix	10	1×	Enzyme activation	95	5 min	1	Ramp rate: 2 °C/s Lid temperature: 105 °C Set the sample volume to 40 μL
Forward primer (2 μM)	1	100 nM	Denaturation	95	30 s	40	
Reverse primer (2 μM)	1	100 nM	Annealing and extension	60	1 min	40	
Diluted cDNA template	2	100 ng/μL	Signal stabilization	4	5 min	1	
DNase-free water	6	—		90	5 min	1	
Total volume	20	—	Hold	4	Infinity	1	

**Table 3 genes-12-00464-t003:** Growth index of *C. medinalis* after dsRNA injection

Injection Treatment	Total Excretion in 7 Days (g)	Weight on Day 7 (g)	Pupal Weight (g)	Pupal Calendar Period (d)	Eclosion Rate (%)	Adult Life Span (d)	Egg Production	Hatching Rate of Eggs (%)
ds*CmUAP*	1.79 ± 0.03 a	0.152 ± 0.03 a	0.144 ± 0.08 a	9.2 ± 0.6 b	58.7 ± 8.5 a	9.8 ± 0.4 a	145.2 ± 12.9 a	68.6 ± 3 a
ds*GFP*	2.25 ± 0.17 b	0.199 ± 0.03 b	0.202 ± 0.11 b	8.2 ± 0.6 b	86 ± 5 b	11.1 ± 0.8 a	86.3 ± 18.1 b	86.2 ± 1.7 b

Note: Data are indicated as the mean ± SD. Different letters in the same columnindicated significant differences (*p* < 0.05, Duncan’s test).

## Data Availability

Data available in a publicly accessible repository.

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
