# Peer review of "Cloning, Characterization, and RNA Interference Effect of the UDP-N-Acetylglucosamine Pyrophosphorylase Gene in Cnaphalocrocis medinalis"

_genes, 2021, doi:10.3390/genes12040464_

Round 1

Reviewer 1 Report

Cloning, Characterization, and RNA interference Effect of the UDP-N-acetylglucosamine pyrophosphorylase Gene in Cnaphalocrocis medinalis

Yuan-Jin Zhou, Juan Du, Shang-Wei Li*, Muhammad Shakeel, Jia-Jing Li, Xiao-Gui Meng

The authors present a well written body of research regarding the use of injected dsRNA homologous to the CmUAP gene to induce RNAi.  The authors also present for the first time a brief characterisation of the CmUAP gene.

Minor revision is required.

Please attend to the following:

Line 68. Delete ‘technology’

Line 69. Delete ‘of’

Line 86. Delete ‘said’

Line 88-89 Replace ‘for the next experiments….’ With. ’until required’

Line 113 Replace ‘Amplify’ with ‘Amplification of….’

Line 142 – 144. Rephrase to make it more clearer what you did here.

Line 161. Change the title here. Presently, It is not representative of the information contained in the section. Eg CmUAP dsRNA synthesis and injection.

Line 116. Replace ‘contained’ with ‘containing’

Line 175 addition of ‘and’ in ‘…EDTA was added ‘..’ the reaction….’

Line 186. I hope you added ‘Petroleum jelly’.

Line 186. Replace ‘inoculated’ with ‘placed’

Line 314. Suspect the ‘e’ picture was an ‘adult injected with dsGFP’

Line 327. Delete ‘are’ and replace ‘consisted’ with ‘consists’

Line 339 – 340 replace ‘Provides a…’ delete ‘as’

Line 344 ‘a studies’…. replace ‘a’ with ‘None’ or ‘limited’

Line 346 ‘…CmUAP ‘is’ highly…..’

Line 347. Italics for the name

Line 349. ‘This investigation….’ ….is ‘evolutionarily’ highly……’ delete ‘evolution’

Line 350. Replace ‘crucial need’ with ‘possible’

Line 351. ……population’s’

Line 353. ‘….levels ‘were’ observed in adult’s’…..’

Line 399.  Reference list is not as per journal requirements.

Author Response

Cover Letter to Reviewer(genes-1147788)

Dear Reviewer,

 We appreciate the time and effort you put into providing feedback on our manuscripts and thank you for your scientifically rigorous and pertinent suggestions and valuable improvements to our papers. We have incorporated all the suggestions you have made. We spent a lot of time making many changes in response to your proposed revisions, including language style, logic, formatting, and argument.These changes are highlighted in the manuscript. Please see the point-by-point responses in red below.

Point 1:Line 68. Delete ‘technology’.

Response 1:Yes, this word is redundant and we have removed it. Thanks.

Point 2:Line 69. Delete ‘of’.

Response 2:We have removed it.

Point 3:Line 86. Delete ‘said’.

Response 3:This word is redundant and we have removed it.

Point 4:Line 88-89 Replace ‘for the next experiments….’ With. ’until required’.

Response 4:We have replaced ‘for the next experiments….’ With. ’until required’,displayed on line  89-90.

Point 5:Line 113 Replace ‘Amplify’ with ‘Amplification of….’.

Response 5:We have replaced ‘Amplify’ with ‘Amplification of….’displayed on line  114-115.

Point 6:Line 142 – 144. Rephrase to make it more clearer what you did here.

Response 6:We have newly defined 'ddPCR ' and rewritten the later inaccurate expressions.on line  143-144

Point 7:Line 161. Change the title here. Presently, It is not representative of the information contained in the section. Eg CmUAP dsRNA synthesis and injection.

Response 7:Yes, the title here doesn't quite sum up the content of this section,we have modified the title  to ‘dsRNA synthesis and injection'.

Point 8:Line 116. Replace ‘contained’ with ‘containing’

Response 8:We replaced 'contained' with 'containing' on line 117

Point 9:Line 175 addition of ‘and’ in ‘…EDTA was added ‘..’ the reaction….’

Response 9:We have rewritten the sentence to ‘ 4 μL EDTA was added to the reaction mixture and kept at 65 °C for 10 min to terminate the reaction’,on line 176-177

Point 10:Line 186. I hope you added ‘Petroleum jelly’.

Response 10:We added ‘Petroleum jelly’,the complete expression is:‘1 μL of dsCmUAP and dsGFP (control) at a concentration of 2 μg/μL was injected with a microinjector in the direction of blood flow and apply petroleum jelly to the wound ’.On  line 185-187

Point 11:Line 186. Replace ‘inoculated’ with ‘placed’

Response 11:We have  replaced ‘inoculated’ with ‘placed’

Point 12:Line 314. Suspect the ‘e’ picture was an ‘adult injected with dsGFP’

Response 12:Thank you for pointing out such a critical error, we have fixed it and it is indeed "adult injected with dsCmGFP"

Point 13:Line 327. Delete ‘are’ and replace ‘consisted’ with ‘consists’

Response 13:Yes, we have fixed the incorrect wording

Point 14:Line 339 – 340 replace ‘Provides a…’ delete ‘as’

Response 14:We have corrected the incorrect wording, thank you for pointing it out

Point 15:Line 344 ‘a studies’…. replace ‘a’ with ‘None’ or ‘limited’

Response 15:We apologize for the incorrect wording and have corrected it

Point 16:Line 346 ‘…CmUAP ‘is’ highly…..’

Response 16:We added the verb "IS"

Point 17:Line 347. Italics for the name

Response 17:We have modified the format

Point 18:Line 349. ‘This investigation….’ ….is ‘evolutionarily’ highly……’ delete ‘evolution’

Response 18:We removed   'evolution' 

Point 19:Line 350. Replace ‘crucial need’ with ‘possible’

Response 19:We have rewritten the sentence as:‘Therefore, UAP genes can be used as target loci to develop broad-spectrum control strategies for lepidopteran insects.’

Point 20:Line 351. ……population’s’

Response 20:We have rewritten the sentence,on line 354-355

Point 21:Line 353. ‘….levels ‘were’ observed in adult’s’…..’

Response 21:We have made changes according to your suggestions on line 357

Point 22:Line 399.  Reference list is not as per journal requirements.

Response 22:We modified the reference list using the "EndNote" package

Thank you again for your valuable comments!

Best regards,

Yuan-jin Zhou , Shang-Wei Li.

 Provincial Key Laboratory for Agricultural Pest Management of Mountainous Regions, Institute of Entomology, Guizhou University, Guiyang, Guizhou 550025, China

Reviewer 2 Report

This paper describes the expression and knockdown via RNAi of a chitin biosynthesis gene, UAP, in the Lepidopteran rice pest, RFL. Overall, the authors did a great job characterizing UAP, finding expression patterns across life stages and tissues, and designing dsRNA that is unique to RFL. It was obvious that the effects of RNAi increased mortality and caused various deformities in RFL larvae and adults. I have very few criticisms of the paper, but my thoughts on improvement are below.

First, the abstract is very data heavy. It would be nice to have a little background about RFL and why we should be interested in learning about it. 

Please be careful of verb tense and other grammatical errors. I also noticed that genus and species names were not italicized as they should be throughout the paper (line 347 and section 3.2). I also noticed the use of the term “insect-pests” when it should be “insect pest”.

Line 60, I did not see PM defined in the text until the discussion.

I’m curious about the adult wing deformities. There is clearly an issue based on the pictures, but I wonder if it is due to the injection process itself and not the dsRNA. I wonder this because the expression level in wing was found to be very low. A little argument about this in the discussion would be appreciated. 

Author Response

Cover Letter to Reviewer (genes-1147788)

Dear Reviewer,

Thank you for your careful and conscientious suggestions on our manuscript, we have made changes to address all your suggestions.These changes are highlighted in the manuscript. Please see the point-by-point responses in red below.

Point 1:First, the abstract is very data heavy. It would be nice to have a little background about RFL and why we should be interested in learning about it.

Response 1:Yes, thank you for the suggestion. We have included a short background on Cnaphalocrocis medinalis in the abstract on line 9-10.We added‘it poses a major threat to rice production and is difficult to control’

Point 2:Please be careful of verb tense and other grammatical errors.

Response 2:We apologize for the bad grammar and have done our best to fix it. For example:we have replaced ‘Amplify’ with ‘Amplification of….’displayed on line  114-115ï¼›

we replaced 'contained' with 'containing' on line 117;

we have  replaced ‘inoculated’ with ‘placed’;

we have delete ‘are’ and replace ‘consisted’ with ‘consists;

……

We have fixed a large number of grammatical and tense errors, as detailed in the attached red revision

Point 3:I also noticed that genus and species names were not italicized as they should be throughout the paper (line 347 and section 3.2).

Response 3:We double-checked species and genus names and words indicating gene names throughout, setting the formatting to italics.Of course, we did not italicize words indicating an enzyme in some cases. Thank you for your meticulous checking!

Point 4:I also noticed the use of the term “insect-pests” when it should be “insect pest”.

Response 4:We replaced 'insect-pests' with 'insect pest' on line 44

Point 5:Line 60, I did not see PM defined in the text until the discussion.

Response 5:We have defined 'PM' in line 62, thank you for your kind reminder

Point 6:I’m curious about the adult wing deformities. There is clearly an issue based on the pictures, but I wonder if it is due to the injection process itself and not the dsRNA. I wonder this because the expression level in wing was found to be very low. A little argument about this in the discussion would be appreciated.

Response 6:We do not have an adequate explanation for the wing changes of insects. We added the following sentence on lines 383-388:We speculate that due to the low expression of CmUAP and reduced synthesis of Chitin, the insects do not have sufficient supplies during the pupal stage when metamorphosis and drastic changes occur, and that the efficiency of RNA interference during the pupal stage is as high as 53.4%(Figure 7), suggesting that CmUAP has a crucial role during the transition from the pupal to the adult stage of RLF, resulting in incomplete and malformed adult wing development.    Because the control group wings were phenotypically normal, we are confident that the appearance of malformed wings was due to RNA interference.

Thank you again for your valuable comments!

Best regards,

Yuan-jin Zhou , Shang-Wei Li.

 Provincial Key Laboratory for Agricultural Pest Management of Mountainous Regions, Institute of Entomology, Guizhou University, Guiyang, Guizhou 550025, China
